# Estimated quantity of swine virus genomes based on quantitative PCR analysis in spray-dried porcine plasma samples collected from multiple manufacturing plants

**Elena Blázquez**[1,2,3], **Joan Pujols**[1,3], **Joaquim Segalés**[3,4,5], **Carmen Rodríguez**[2], **Joy Campbell**[6], **Louis Russell**[6], **Javier Polo**[2,6]*

**1** IRTA, Centre de Recerca en Sanitat Animal (CReSA-IRTA), Bellaterra, Barcelona, Spain, **2** APC EUROPE S.L.U., Granollers, Barcelona, Spain, **3** OIE Collaborating Centre for the Research and Control of Emerging and Reemerging Swine Diseases in Europe (IRTA-CReSA), Bellaterra, Barcelona, Spain, **4** Departament de Sanitat i Anatomia Animals, Universitat Autònoma de Barcelona (UAB), Bellaterra, Barcelona, Spain, **5** UAB, Centre de Recerca en Sanitat Animal (CReSA, IRTA-UAB), Campus de la Universitat Autònoma de Barcelona, Bellaterra, Barcelona, Spain, **6** APC LLC, Ankeny, Iowa, United States of America

* javier.polo@apc-europe.com

**Data Availability Statement:** All relevant data are within the paper and its Supporting information files.

## Abstract

This survey was conducted to estimate the incidence and level of potential viral contamination in commercially collected porcine plasma. Samples of spray dried porcine plasma (SDPP) were collected over a 12- month period from eight spray drying facilities in Spain, England, Northern Ireland, Brazil, Canada, and the United States. In this survey, viral load for several porcine pathogens including SVA, TGEV, PRRSV (EU and US strains), PEDV, PCV-2, SIV, SDCoV and PPV were determined by qPCR. Regression of Ct on $TCID_{50}$ of serial diluted stock solution of each virus allowed the estimate of potential viral level in SDPP and unprocessed liquid plasma (using typical solids content of commercially collected porcine plasma). In this survey SVA, TGEV or SDCoV were not detected in any of the SDPP samples. Brazil SDPP samples were free of PRRSV and PEDV. Samples of SDPP from North America primarily contained the PRRSV-US strain while the European samples contained the PRRSV-EU strain (except for one sample from each region containing a relatively low estimated level of the alternative PRRSV strain). Estimated viral level tended to be in the range from <1.0 $\log_{10}$ $TCID_{50}$ to <2.5 $\log_{10}$ $TCID_{50}$. Estimated level of SIV was the exception with a very low incidence rate but higher estimated viral load <3.9 $\log_{10}$ $TCID_{50}$. In summary, the incidence of potential viral contamination in commercially collected porcine plasma was variable and estimated virus level in samples containing viral DNA/RNA was relatively low compared with that occurring at the peak viremia during an infection for all viruses or when considering the minimal infectious dose for each of them.

**Funding:** Funding for this study was provided by APC Europe, S.L.U., Granollers, Spain, and APC LLC, Ankeny, IA, 50021, USA. These companies manufacture animal blood products for animal consumption. The funders provided support in the form of salaries for authors EB, CR, JC, LR and JPolo, but did not have any additional role in the study design, data collection and analysis, decision to publish, or preparation of the manuscript. The specific roles of these authors are articulated in the 'author contributions' section.

**Competing interests:** The authors have read the journal's policy and the authors of this manuscript have the following competing interests: EB, CR, and JPolo are employed by APC Europe, S.L.U. Granollers, Spain and JC, LR and JPolo are employed by APC LLC, Ankeny, IA, USA. APC Europe and APC LLC manufactures and sells spray-dried animal plasma; however, the companies did not have any additional role in the study design, data collection and analysis, decision to publish, or preparation of the manuscript. This does not alter the authors' adherence to all PLOS ONE policies on sharing data and materials. JPujols, and JS declared no conflict of interest.

## Introduction

Spray dried porcine plasma (SDPP) is a complex mixture of functional components including immunoglobulins, albumin, transferrin, fibrinogen, lipids, growth factors, bioactive peptides, enzymes, hormones, and amino acids commonly used in feed for young animals including pigs, calves, and poultry [1–4].

It has been speculated that the use of SDPP in swine feed contributed to the spread of infective viruses such as *Porcine circovirus 2* (PCV-2) and *Porcine epidemic diarrhea virus* (PEDV) [5–7]. However, other evidence demonstrates that reduced mortality and morbidity is associated with the use of SDPP in pig diets [1, 3, 8, 9] and experimental and epidemiological evidence demonstrate that SDPP does not spread diseases [10–12].

The manufacturing process to produce SDPP includes multiple hurdles steps that have been validated to inactivate potential viral contamination. These hurdles include spray drying (SD, 80˚C throughout substance), ultraviolet light (UV) treatment (3000 J/L) and post drying storage (PDS) at 20˚C for 14 d [13–19]. Depending on the virus, the theoretical cumulative inactivation for SD and PDS range from 5.8 to 9.1 $\log_{10}$ $TCID_{50}$/g liquid plasma, while SD, PDS and UV range from 11.7 to 20.9 $\log_{10}$ $TCID_{50}$/g liquid plasma (Table 1). The World Health Organization recommends cumulative robust inactivation procedures capable of inactivating 4 $\log_{10}$ of virus by each of these steps in the manufacturing process for human blood and plasma products [20, 21].

While the inactivation capacity of the multiple hurdle manufacturing process has been validated for several economically important swine viruses, it is also important to estimate the potential virus quantity in liquid plasma used to produce SDPP. Therefore, this survey was conducted to estimate the quantity and determine the frequency of genome detection of different swine viruses in commercially produced SDPP samples collected from 8 different manufacturing plants. Results obtained from quantitative polymerase chain reaction (qPCR) analyses of the SDPP samples were used to infer the potential viral contamination in the liquid porcine plasma from which it was produced.

**Table 1. Different inactivation steps involved in the manufacturing process of spray dried porcine plasma.** Inactivation expressed as $\log_{10}$ reduction values (LRVs) $TCID_{50}$/g for viruses.

| Virus Type | | | Spray-Drying | UV-C* | Storage at 20˚C for 14 d | Combined Theoretical Inactivation | References |
|---|---|---|---|---|---|---|---|
| RNA | Enveloped | Porcine reproductive and respiratory syndrome virus (PRRSV) | >4.0 | 12.9 ± 0.3 | >4.0 | >20.9 | [13, 17, 62] |
| | | Swine influenza virus (SIV) | 2.8** | ± 0.2 | 3.2** | 13.9 | [17] |
| | | Porcine epidemic diarrhea virus (PEDV) | 5.1 4.2 | 6.6 ± 0.1 | 3.8 | 14.6–15.5 | [15–17] |
| | | Classical swine fever virus (CSFV) | 5.8 | 7.9 ± 0.2 | ND | >13.7 | [17, 63] |
| | Naked | Swine vesicular disease virus (SVDV) | 6.7 | 3.5 ± 0.07 | ND | >10.2 | [14, 17] |
| | | Senecavirus A (SVA) | ND | 4.0 ± 0.08 | >5.0** | >9.0 | [17] |
| DNA | Enveloped | Pseudorabies virus (PRV) | 5.3 | 8.1 ± 0.2 | ND | >13.4 | [13, 17] |
| | | African swine fever virus (ASFV) | 4.1 ± 0.2 | 6.8 ± 0.1 | >5.7 | >16.6 | [17, 19, 63] |
| | Naked | Porcine parvovirus (PPV) | 2.7** | 6.0 ± 0.1 | 3.1** | >11.8 | [17] |

LRVs with symbol > results indicate the inactivated amount in the processed sample exceeded the amount inoculated in the initial sample before processing or storage.
[1]ND = Not determined.
*The UV log-kill estimated values were calculated commercial UV dosage (3251 J/L) by the estimated D-value from Blázquez et al., [17].
**University of Minnesota. Understanding the risk of virus transmission in spray dried porcine plasma–food safety assessment. 2020. Unpublished data.

## Material and methods

### Ethical statement

No animals were used for the study conducted.

### Spray-dried porcine plasma sample collection

One sample per month was collected from a randomly selected commercial lot of SDPP during 12 consecutive months from eight different manufacturing plants located in Iowa, USA (IA-USA), North Carolina, USA (NC-USA), Santa Catarina, Brazil (SC-Brazil), central Spain (C-Spain), northeastern Spain (NE-Spain), central England (C-England) and Northern Ireland (N-Ireland). The N-Ireland manufacturing plant collects porcine blood from abattoirs located both, in Republic of Ireland and Northern Ireland. Samples from a manufacturing plant located in Quebec, Canada (QB-Canada), were taken biweekly during a 6 month-period.

Samples were collected from July 2018 to June 2019 (SC-Brazil), August 2018 to July 2019 (IA-USA, NE-Spain, C-Spain and N-Ireland) or September 2018 to August 2019 (NC-USA, C-England). The QB-Canada plant provided 12 samples randomly collected from March to August 2019. The collected SDPP samples represented a single point in time, not the entire month. Whole blood or plasma was chilled and stored in insulated agitated tanks at the abattoir. transported to the spray drying facility in dedicated tankers and stored and may be blended with plasma from different slaughterhouses in agitated silos before drying. In the manufacturing plants used in this study, a manufacturing lot of SDPP can range between 3,000 to 15,000 kg of plasma depending on the plant. Therefore, one lot of SDPP represented between 16,650 to 166,500 pigs. During the 12-month collection period, samples were stored in whirl packs (Whirl-Pak®, Nasco, Madison, WI) and held at each plant in the quality assurance laboratory (room temperature) during the collection period. Subsequently, all SDPP samples were sent to the IRTA-CReSA Animal Health Research Center in Barcelona, Spain, and stored (-20°C) until analyses for virus genome. One sample collected in December from the IA-USA plant was damaged during transport and was not used for analysis. Therefore, a total of 95 SDPP samples were analyzed.

### Sample analysis by PCR

All SDPP samples were re-solubilized in distilled water at the ratio 1:9 of SDPP: water volume to represent the typical solid content in liquid plasma. Two hundred milliliters of diluted plasma sample were used for nucleic acid extraction using MagMAX™ Pathogen RNA/DNA Kit (Thermo Fisher Scientific, MA, USA). The recommended quantity of purified nucleic acids was amplified using real time PCR kits for PCV-2 (LSI VetMAX™ Porcine Circovirus Type 2 Quantification, Thermo Fisher Scientific, MA, USA), *Porcine reproductive and respiratory syndrome virus* [PRRSV] European and North American strains (LSI VetMAX™ PRRSV EU/NA Real-Time PCR Kit; Thermo Fisher Scientific, MA, USA), *Swine influenza virus* [SIV] (EXOone Influenza A, EXOPOL, Zaragoza, Spain), *Porcine parvovirus* [PPV] (VetMAX™ Porcine Parvovirus Kit, Thermo Fisher Scientific, MA, USA), PEDV, *Transmissible gastroenteritis virus* [TGEV] and *Swine deltacoronavirus* [SDCoV] (VetMAX™ PEDV/TGEV/SDCoV, Thermo Fisher Scientific, MA, USA) and *Senecavirus A* [SVA] (EXOone Seneca Virus Valley, EXOPOL, Zaragoza, Spain).

According to all PCR kit guidelines, virus genome results with Ct values >40 were considered negative.

## Virus stock production for development of standard curves to convert PCR Ct to $TCID_{50}$/g SDPP

From those viruses detected in SDPP by qPCR, a stock of each virus was produced in the laboratory. Seven serial dilutions of viral stocks (PEDV, PRRSV-1 (EU strain), PRRSV-2 (US strain), PPV-1, PCV-2 and SIV) were analyzed by quantitative PCR/RT-PCR (obtaining the corresponding Ct value) and $TCID_{50}$ titration. Standard curves were established for each virus by regressing $TCID_{50}$/g SDPP on Ct results [Fig 1]. Those viral stocks were used as an internal standard on each amplification run/plate and quantitative PCR/RT-PCR Ct values extrapolated to $TCID_{50}$. Potential viral quantity determined on SDPP was corrected for typical solids content for each commercially collected plasma. $TCID_{50}$ titers were calculated by the Reed and Muench method [22].

**Porcine reproductive and respiratory syndrome virus.** *Porcine reproductive and respiratory syndrome virus* 3268 EU strain was propagated in porcine alveolar macrophages (PAM) grown in standard growth media (SGM) containing minimum essential medium eagle (MEM-E; ThermoFisher, Waltham, MA, USA) supplemented with 1% penicillin 10,000 U/mL and streptomycin 10 mg/mL (ThermoFisher), 0.5% Nystatin 10,000 IU/mL (Sigma-Aldrich, Burlington, MA, USA), 1% L-glutamine 200 mM (ThermoFisher) plus 5% fetal bovine serum (FBS). Cells were cultured in 75-cm$^2$ flasks. When cells were confluent, the media was discarded, and the adsorption was done using the virus at 0.01 multiplicity of infection (MOI). After 1.5 hours at 37ºC, inoculum was removed, and 30 mL of medium were added. Titration was done in triplicate obtaining a final titer of $10^{5.5\pm0.2}$ $TCID_{50}$/mL.

*Porcine reproductive and respiratory syndrome virus* RV2332 US strain was propagated in MARC145 cells (ATCC No. CRL-12231) (kindly provided by Dr. Enric Mateu, *Universitat Autònoma de Barcelona*, Barcelona, Spain) using SGM supplemented with 10% FBS as explained above until a viral stock solution with a final titer of $10^{4.9\pm0.4}$ $TCID_{50}$/mL was obtained.

**Porcine epidemic diarrhea virus.** *Porcine epidemic diarrhea virus* CV777 strain [23], kindly provided by Dr. Hans Nauwynck (University of Ghent, Belgium), was propagated in VERO cells (ATCC CCL-81) grown in SGM with 10% FBS. Cells were cultured in 175-cm$^2$ flask and when they were confluent, the media was removed, and cells were rinsed twice with phosphate buffered saline (PBS). Finally, inoculum was added at 0.001 MOI and adsorption was done for 1 hour at 37ºC. Subsequently, the inoculum was discarded, flasks were rinsed twice with PBS and SGM supplemented with 10 mg/mL trypsin, and 0.3% tryptose (Sigma-Aldrich, Burlington, MA, USA). The viral stock was produced in the same cells and was titrated in triplicate obtaining a suspension with a viral titer of $10^{5.4\pm0.1}$ $TCID_{50}$ /mL.

**Swine influenza virus.** *Swine influenza virus* strain H1N1 A/Swine/Spain/SF11131/2017 [24] was propagated in MDCK cell line (ATCC CCL-34) grown in DMEM (ThermoFisher, Waltham, MA, USA) supplemented with 1% penicillin (10,000 U/mL), 1% streptomycin (10 mg/mL; ThermoFisher), 0.5% Nystatin (10,000 U/mL) (Sigma-Aldrich, Burlington, MA, USA), 1% L-glutamine 200mM (ThermoFisher) and 5% FBS. Cells were cultured in 175-cm$^2$ flask. When cells were confluent, the media was discarded, and the adsorption was done at 0.1 MOI. After 1 hour at 37ºC, inoculum was removed, and 30 mL of medium were added. The viral suspension was titrated in triplicate and the final virus titer was $10^{7.6\pm0.2}$ $TCID_{50}$ /mL.

**Porcine circovirus 2.** Porcine circovirus 2 genotype b isolate Sp-10-7-54-13 [25] was cultured in the PK-15 cell line (provided by the Institute of Virology UE and OIE Reference Laboratory for CSFV, Hannover), grown in SGM with 10% FBS. A mix of 6 mL of virus stock and 7 x $10^6$ PK-15 cells resuspended in 50 mL of MEM-E (MOI 0.1) were added in 175 and 25 cm$^2$ flasks. At 24 hours cells were treated with glucosamine (Sigma-Aldrich, Burlington, MA, USA)

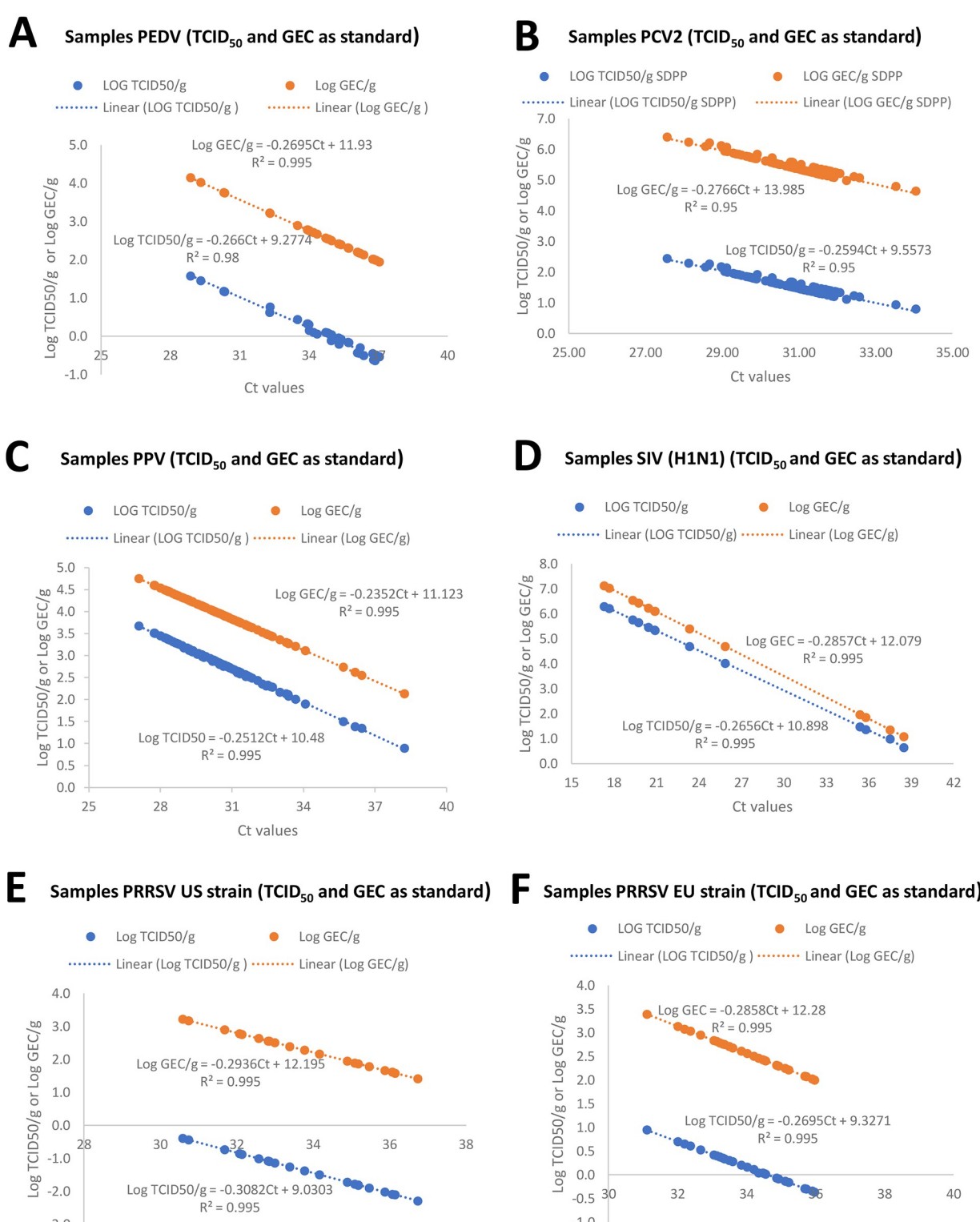

**Fig 1. Regression curves between Ct values and tissue culture infectious dose 50 (TCID$_{50}$/g) or Genome equivalent copies (GEC/g) of spray-dried porcine plasma (SDPP).** Values expressed in log$_{10}$ TCID$_{50}$/g SDPP or log$_{10}$ GEC/g SDPP. Each box includes the spot values of the SDPP samples analyzed and the regression equation between Ct and TCID$_{50}$/g or GEC/g SDPP and the r$^2$ value. A.Regression curves for *porcine epidemic diarrhea virus* (PEDV); B. Regression curves for *porcine circovirus type-2* (PCV-2); C. Regression curves for *porcine parvovirus* (PPV); D. Regression curves for *swine influenza virus* (SVI) H1N1; E. Regression curves for *porcine reproductive and respiratory syndrome virus* (PRRSV) US strain; F. Regression curves for PRRSV EU strain.

to facilitate the virus infection. Forty-eight hours later, viral infection was checked by immunoperoxidase monolayer assay (IPMA) [26] in the 25 $cm^2$ flask. If more than 25 positive cells were counted in a microscope field, the 175 $cm^2$ flask was trypsinized and the cells were transferred to 3 new 175 $cm^2$ flasks. The virus stock was titrated in triplicate with a final titer of $10^{5.5\pm0.04}$ $TCID_{50}$ /mL.

**Porcine parvovirus.** Porcine parvovirus strain NADL-2 was kindly provided by Dr Albert Bosch (Department of Genetics, Microbiology and Statistics School of Biology, University of Barcelona, Spain). It was propagated in SK-RST cells (ATCC CRL-2842), grown in SGM supplemented with 5% FBS. One mL of virus stock and 9 mL of MEM-E supplemented with 1% pyruvate (Merck KGaA, Darmstadt, Germany) were added to a conical tube with 16 x $10^6$ SK-6 cells and shaken for 30 minutes at 104 rpm and 37ºC. After that time, the contents of the tube were transferred to a 175 $cm^2$ flask, in which 40 mL of MEM-E supplemented with 1% pyruvate were added. Inoculated flasks were incubated for four days at 37ºC until CPE was observed. A viral suspension was obtained and titrated in triplicate, obtaining a final viral solution of $10^{6.6\pm0.2}$ $TCID_{50}$ /mL.

## Estimation of $TCID_{50}$ and genomic equivalent copies (GEC) from Ct values obtained from q-PCR results

To establish equivalence of positive qPCR results (measured as Ct values) with $TCID_{50}$/mL and viral genome equivalent copies (GEC) content, seven serial dilutions of abovementioned titrated virus stocks were performed, and virus genome amplified with a second set of PCR kits (GPS, Genetic PCR Solutions Alicante, Spain). Each kit contained a genome quantified standard for the different viruses tested: PRRSV (PRRSV-I dtec-RT-qPCR, PRRSV-II dtec-RT-qPCR), PEDV (PEDV dtec-RT-qPCR), PPV (PPV-1 dtec-RT-qPCR) and SIV (SIV dtec-RT-qPCR).

## Statistical analysis

Dilutions of titrated viral stocks were included as an internal standard on each amplification PCR run containing SDPP samples. The Excel software was used to obtain the equation correlating $TCID_{50}$ and Ct values as well as GEC and Ct values. Then, results of the different PCR techniques originally expressed as Ct values for each SDPP sample tested were extrapolated to virus infectious particles and GEC based on the obtained regression formulae.

Average, number of observations, standard deviation, minimum value, maximum value, and ranges were calculated within each virus and for each SDPP producing plant using LSMEANS (SAS 9.4, 2016).

## Results and discussion

In this survey, viral loads for several porcine pathogens including SVA, TGEV, PRRSV (EU and US strains), PEDV, PCV-2, SIV, SDCoV and PPV were determined by qPCR in reconstituted commercial SDPP. First, the Ct values from serial dilutions of a stock solution for each virus allowed the development of a regression equation between Ct and $TCID_{50}$ that allowed an estimate of the viral titers in the SDPP samples. Finally, using typical solids content of unprocessed liquid plasma, the viral level in liquid plasma was adjusted per gram ($TCID_{50}$/g liquid plasma). The relationships between Ct and $TCID_{50}$ of serial diluted stock solutions were linear with a correlation coefficient from 0.95 to 0.995 (Fig 1). Similar correlation coefficients were found when regressing Ct on $\log_{10}$ GEC/g on the tested samples (Fig 1). The slope of the lines for either $TCID_{50}$ or GEC/g were similar, while the intercepts were different (Fig 1), consistent with the fact that not all viral genome copies are infective [27]. There was variability

between infectious particles and genome copy numbers observed among tested viruses, with less than 1 log difference for SIV to around 4 log differences for PCV-2.

Previous research has shown PCR/RT-PCR Ct values in SDPP to be relatively stable during normal storage conditions [19, 28, 29]. Similar levels of viral genome were detected in plasma inoculated with PCV-2 or SIV before and after spray drying (E. Blázquez, personal communication). The stability of PCR Ct values, the linear relationship between Ct and $TCID_{50}$ and the linear relationship between Ct and GEC provides additional assurance that estimated viral contamination of commercially collected SDPP and estimates of liquid plasma are accurate.

Frequency of detection and estimated quantity of virus in SDPP samples mimicking unprocessed liquid plasma samples collected at different plants is presented in Tables 2 and 3.

The S1 Table -SDPP includes monthly (during the years 2018–2019) Ct values and estimated virus levels reported as $log_{10}$ GEC/g and $log_{10}$ $TCID_{50}$/g in reconstituted SDPP from the different manufacturing plants located in different swine production areas around the world.

**Table 2. Ct values and estimated viral genome presence expressed in $log_{10}$ genome equivalent copies (GEC) and $log_{10}$ $TCID_{50}$/g spray dried porcine plasma in manufacturing plants located in different swine production areas around the world during the years 2018–2019.** Values expressed as Average ± SD for positive samples.

| Plant | US-IA (n = 11) | US-NC (n = 12) | Canada (n = 12) | Spain-NE (n = 12) | Spain-C (n = 12) | England (n = 12) | NI (n = 12) | Brazil (n = 12) |
|---|---|---|---|---|---|---|---|---|
| **PEDV** | | | | | | | | |
| Ct | 33 ± 3 | 34 ± 2 | 34 | 35 ± 1 | 35 ± 1 | Neg | Neg | Neg |
| $log_{10}$ GEC/g | 2.9 ± 0.9 | 2.7 ± 0.6 | 2.7 | 2.4 ± 0.3 | 2.4 ± 0.4 | | | |
| $log_{10}$ $TCID_{50}$/g | 0.3 ± 0.9 | 0.1 ± 0.6 | 0.3 | 0.01 ± 0.33 | -0.05 ± 0.38 | | | |
| % Positive samples | 82 | 50 | 8 | 83 | 67 | 0 | 0 | 0 |
| **PCV-2** | | | | | | | | |
| Ct | 32 ± 1 | 31 ± 2 | 30 ± 1 | 30 ± 1 | 30 ± 1 | 31 ± 1 | 31 ± 1 | 31.0 ± 0.4 |
| $log_{10}$ GEC/g | 5.3 ± 0.2 | 5.5 ± 0.5 | 5.7 ± 0.3 | 5.5 ± 0.2 | 5.6 ± 0.3 | 5.4 ± 0.4 | 5.4 ± 0.2 | 5.3 ± 0.1 |
| $log_{10}$ $TCID_{50}$/g | 1.4 ± 0.2 | 1.6 ± 0.5 | 1.8 ± 0.3 | 1.6 ± 0.2 | 1.7 ± 0.3 | 1.5 ± 0.4 | 1.5 ± 0.2 | 1.4 ± 0.1 |
| % Positive samples | 100 | 100 | 100 | 100 | 100 | 100 | 100 | 100 |
| **PPV** | | | | | | | | |
| Ct | 30 ± 1 | 32 ± 2 | 31 ± 1 | 31 ± 3 | 31 ± 1 | 30 ± 1 | 28.4 ± 0.5 | 31 ± 1 |
| $log_{10}$ GEC/g | 4.0 ± 0.3 | 3.5 ± 0.6 | 3.9 ± 0.3 | 3.9 ± 0.8 | 3.9 ± 0.3 | 4.0 ± 0.3 | 4.4 ± 0.1 | 3.8 ± 0.2 |
| $log_{10}$ $TCID_{50}$/g | 2.8 ± 0.3 | 2.4 ± 0.6 | 2.8 ± 0.4 | 2.7 ± 0.8 | 2.8 ± 0.3 | 2.9 ± 0.3 | 3.3 ± 0.1 | 2.6 ± 0.3 |
| % Positive samples | 100 | 100 | 100 | 100 | 100 | 100 | 100 | 100 |
| **SIV** | | | | | | | | |
| Ct | 38 | Neg | 35 | 23 ± 4 | 19.6 ± 0.3 | 24 ± 11 | 21 | 28 ± 10 |
| $log_{10}$ GEC/g | | | | | | | | |
| $log_{10}$ $TCID_{50}$/g | -1.3 | | 0.4 | 3.9 ± 1.1 | 5.0 ± 0.1 | 3.8 ± 3.0 | 4.6 | 2.7 ± 2.7 |
| % Positive samples | 9 | 0 | 8 | 17 | 17 | 25 | 8 | 25 |
| **PRRS-US** | | | | | | | | |
| Ct | 33 ± 2 | 34 ± 1 | 34 ± 2 | Neg | 36 | Neg | Neg | Neg |
| $log_{10}$ GEC/g | 2.4 ± 0.5 | 2.1 ± 0.4 | 2.2 ± 0.7 | | 1.6 | | | |
| $log_{10}$ $TCID_{50}$/g | -1.3 ± 0.5 | -1.5 ± 0.4 | -1.5 ± 0.7 | | -2.1 | | | |
| % Positive samples | 100 | 17 | 50 | 0 | 8 | 0 | 0 | 0 |
| **PRRS-EU** | | | | | | | | |
| Ct | 36 | Neg | Neg | 35 ± 1 | 34 ± 2 | 34 ± 1 | 34 ± 1 | Neg |
| $log_{10}$ GEC/g | 2.1 | | | 2.4 ± 0.3 | 2.6 ± 0.5 | 2.7 ± 0.4 | 2.6 ± 0.3 | |
| $log_{10}$ $TCID_{50}$/g | -0.3 | | | 0.03 ± 0.24 | 0.2 ± 0.4 | 0.3 ± 0.4 | 0.2 ± 0.3 | |
| % Positive samples | 9 | 0 | 0 | 33 | 58 | 50 | 83 | 0 |

**Table 3. Estimated quantification of different viruses' genomes expressed in $\log_{10}$ TCID$_{50}$/g ± SD (percentage of positive samples) in unprocessed raw liquid plasma from PCR or RT-PCR analyses of spray dried porcine plasma samples collected at different plants.**

| Plant | PEDV | PCV-2 | PPV | SIV | PRRS- US | PRRS-EU |
|---|---|---|---|---|---|---|
| US-IA | -0.8 ± 0.9 | 0.3 ± 0.2 | 1.7 ± 0.3 | -2.5 | -2.4 ± 0.5 | -1.4 |
| US-NC | -0.9 ± 0.6 | 0.6 ± 0.5 | 1.3 ± 0.6 | Neg | 2.6 ± 0.4 | Neg |
| Canada | -0.8 | 0.6 ± 0.3 | 1.7 ± 0.4 | -0.7 | -2.5 ± 0.7 | Neg |
| Spain-NE | -1.0 ± 0.3 | 0.6 ± 0.2 | 1.7 ± 0.8 | 2.9 ± 1.1 | Neg | -1.0 ± 0.3 |
| Spain-C | -1.2 ± 0.4 | 0.5 ± 0.3 | 1.7 ± 0.3 | 3.8 ± 0.1 | -3.2 | -0.9 ± 0.4 |
| England | Neg | 0.5 ± 0.4 | 1.9 ± 0.3 | 2.8 ± 3 | Neg | -0.8 ± 0.4 |
| Northern Ireland | Neg | 0.4 ± 0.2 | 2.2 ± 0.1 | 3.5 | Neg | -0.9 ± 0.3 |
| Brazil | Neg | 0.3 ± 0.1 | 1.5 ± 0.3 | 1.6 ± 2.7 | Neg | Neg |
| Range | -1.8–0.5 | -0.3–1.4 | -0.2 –-2.6 | -2.5–4.6 | -3.2 –-1.5 | -1.5 –-0.2 |

The S2 Table (Raw Plasma) includes estimated viral levels in unprocessed plasma reported as $\log_{10}$ TCID$_{50}$/g. It is important to recognize that a positive PCR/RT-PCR does not imply infectivity [16], a fact that was observed for all the viruses studied in the present work.

In this survey neither SVA, TGEV nor SDCoV were detected in any of the SDPP samples. SVA infection has been detected in the Americas and Asia, but not in Europe [30]. Viremia and clinical signs in SVA infected pigs appear within 2 to 3 days post-inoculation and last for few days [31, 32]; therefore, there was minimal chance of an infected pig being undetected at the farm or during antemortem inspection. Despite SVA infected animals have been sporadically detected on-farm and at abattoirs during ante-mortem inspection [33], effective identification of farm outbreaks and surveillance system in place probably contributed to the absence of SVA genome in the tested SDPP samples. Further supporting this hypothesis, a US survey reported only 1.2% of oral samples from 25 states being RT-PCR positive for SVA [34]. On the other hand, the inability to detect TGEV in these samples is also consistent with a very low incidence in the US and European swine population [35–37]. In case of SDCoV, the current data agree with prevalence results from Puente et al. [38] that indicated absence of SDCoV and TGEV in 106 Spanish pig farms analyzed between 2017–2019. Furthermore, Ajayi et al. [39] indicated that the presence of SDCoV in Ontario farms decreased from 1.14% in 2014 to 0.08% in 2016, matching with our results of very low presence of SDCoV in the North American pig population analyzed in 2018–19. Noteworthy, samples from Brazil were negative for both PRRSV and PEDV, which is consistent with other reports indicating that these viruses are not present in this country [40–45].

All SDPP samples were tested for both the EU and US strains of PRRSV independently of the geographical origin of the SDPP. Samples from the US contained PRRSV genotype 2, except for one sample from US-IA that had a PRRSV genotype 1 RT-PCR positive result (Ct of 36, equivalent to -0.3 $\log_{10}$ TCID$_{50}$/g SDPP). Similarly, the samples from EU contained the PRRSV genotype 1, except for one sample from Spain-C that had PRRSV genotype 2 positivity (Ct of 36, equivalent to -2.1 $\log_{10}$ TCID$_{50}$/g SDPP). The detection frequency of positive samples differed between plants, with 100% in those from US-IA, 17% in US-NC and 50% in Canada production plants. In Europe, the RT-PCR positivity against PRRSV was 33% for Spain-NE, 58% for Spain-C, 50% for England and 83% for N-Ireland. However, in both the US and in the EU, the estimated PRRSV TCID$_{50}$ in SDPP was < 2 virus particle/g SDPP, with an average Ct of 34 ± 2 and 34 ± 1 for genotype 2 and 1, respectively. Other works have reported low incidence of PRRSV viremia in slaughtered aged pigs [46] and differences in infection prevalence among US geographical areas [47], which is aligned with the results obtained in the present survey.

Estimated PEDV levels in SDPP was <2.0 $\log_{10}$ PEDV/g SDPP. The detection frequency of positive samples was 82% in US-IA, 50% in US-NC and 8% in Canada. These results indicated that PEDV genome distribution was low in Eastern Canada compared with the USA and agrees with surveillance of PEDV cases reported in North America [48, 49]. In Europe, the incidence of positive PEDV samples was 83% in Spain-NE, and 67% in Spain-C while in England and N-Ireland the samples were negative. Although the present study was not designed to elucidate seasonal differences in the estimated quantity for PEDV genome in the different parts of the world, the results suggest a higher frequency of detection and viral loads during the winter, while it was lower in summertime (S1 and S2 Tables). These results are consistent with the observation that PEDV is more stable in cold environments [50] and has a lower incidence of clinical diarrhea cases at farms during the summer season [51].

Both PPV and PCV-2 are stable non-enveloped DNA viruses [52, 53]. Frequency of detection of both PPV and PCV-2 was 100%, since all samples tested positive for genetic material. In all regions, the estimated level of PCV-2 was <2.0 $\log_{10}$ TCID$_{50}$/g SDPP, while PPV presence was <3.0 $\log_{10}$ TCID$_{50}$/g SDPP. Other studies have reported low levels of PCV-2 viremia in finishing swine [54, 55], in part due to the widespread use of PCV-2 vaccine [56, 57]. In addition, PCV-2 infections typically occur during the nursery and growing periods, so, most of animals reach slaughterhouse immunized and with low levels or no circulating virus [58]. On the other hand, PPV vaccines are commonly used in sows globally; considering the duration of PPV maternally derived immunity [53], it was expected to have evidence of natural infection in late finisher pigs. This was confirmed with the present study.

Detection frequency of SIV RNA was very sporadic and the range of potential viral contamination was variable. In IA, NC and Canada, 9%, 0% and 8% of samples yielded positive results, respectively, and estimated amount of viable virus was <1.0 $\log_{10}$ TCID$_{50}$/g SDPP. Similarly, the frequency of detection of SIV in Spain-C, Spain-NE, England, N-Ireland and Brazil was 17%, 17%, 25%, 8% and 25%, respectively. However, when SIV was present, a very wide range of viral loads were obtained, from 0.3 to 5.6 $\log_{10}$ TCID$_{50}$/g SDPP (corresponding to -0.7 to 4.6 $\log_{10}$ TCID$_{50}$/g liquid raw plasma). It is speculated that slower line speed of abattoirs in Europe and Brazil compared to that in US and Canada, resulting in longer time for blood collection that may contribute to increased levels of SIV contamination.

Estimated levels of infectious viruses in commercially collected porcine plasma was significantly lower than viral levels at peak viremia of pigs [31, 46, 56, 59]. Commercially collected porcine plasma is harvested from animals that have been inspected and passed as fit for slaughter for human consumption, precluding collection of blood from clinically sick animals. Typically, market hogs have been vaccinated for many of the economically important diseases and have developed effective immunity [60, 61]. Combined inactivation by multiple hurdles for the viruses analyzed in this study would be >6 $\log_{10}$ TCID$_{50}$/g SDPP for spray drying and post drying storage and >10 $\log_{10}$ TCID$_{50}$/g SDPP if UV-C if also included (Table 1).

In summary, the data from this survey allowed the estimation of potential viral contamination in commercially collected porcine plasma. Estimated level of viral contamination in commercially collected porcine plasma ranged from <2.0 $\log_{10}$ TCID$_{50}$ for most viruses with infrequent SIV levels as high as 4.5 $\log_{10}$ TCID$_{50}$/g liquid plasma. The multiple hurdles in the manufacturing process (UV-C, spray drying and post drying storage) are theoretically capable of inactivating much higher levels of virus (11 to 20 $\log_{10}$ TCID$_{50}$). These data suggest that the multiple hurdles in the manufacturing process of SDPP should be sufficient to inactivate much higher loads of viruses than the potential viral contamination that can be detected in commercially collected porcine plasma.

## Supporting information

**S1 Table. SDPP.** Ct values and estimated virus genome presence in SDPP per months during the years 2018–2019.
(XLSX)

**S2 Table. Raw plasma.** Estimated virus genome presence in raw plasma per months during the years 2018–2019.
(XLSX)

## Acknowledgments

The authors want to appreciate the help provided by the manufacturing and quality assurance staff of all the manufacturing plants involved in this research for their support providing the samples used in this study.

## Author Contributions

**Conceptualization:** Elena Blázquez, Joan Pujols, Joaquim Segalés, Louis Russell.

**Data curation:** Elena Blázquez, Joan Pujols, Javier Polo.

**Formal analysis:** Elena Blázquez, Joan Pujols, Joaquim Segalés, Joy Campbell, Louis Russell, Javier Polo.

**Funding acquisition:** Carmen Rodríguez, Joy Campbell, Louis Russell, Javier Polo.

**Investigation:** Elena Blázquez, Joan Pujols, Joaquim Segalés, Carmen Rodríguez, Joy Campbell, Louis Russell, Javier Polo.

**Methodology:** Elena Blázquez, Joan Pujols, Joaquim Segalés.

**Project administration:** Carmen Rodríguez, Javier Polo.

**Resources:** Elena Blázquez, Javier Polo.

**Software:** Elena Blázquez, Joan Pujols, Joy Campbell.

**Supervision:** Joan Pujols, Joaquim Segalés, Carmen Rodríguez, Joy Campbell, Louis Russell, Javier Polo.

**Validation:** Elena Blázquez, Joan Pujols, Joaquim Segalés, Joy Campbell, Louis Russell, Javier Polo.

**Visualization:** Elena Blázquez, Javier Polo.

**Writing – original draft:** Elena Blázquez, Joan Pujols, Joaquim Segalés, Joy Campbell, Louis Russell, Javier Polo.

**Writing – review & editing:** Elena Blázquez, Joan Pujols, Joaquim Segalés, Carmen Rodríguez, Joy Campbell, Louis Russell, Javier Polo.

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
