## [Decision Letter · Decision Letter 0]

2 Feb 2022

PONE-D-21-33668Estimated quantity of swine virus genomes based on quantitative PCR analysis in spray-dried porcine plasma samples collected from multiple manufacturing plants.PLOS ONE

Dear Dr. Polo Pozo,

Thank you for submitting your manuscript to PLOS ONE. After careful consideration, we feel that it has merit but does not fully meet PLOS ONE’s publication criteria as it currently stands. Therefore, we invite you to submit a revised version of the manuscript that addresses the points raised during the review process.

Please address all comments made by the reviewers. In particular, the use of low, medium, and high is too subjective. Please change all reference to this qualitative result to quantitative results.

Also, please differentiate if Table 1 is log TCID50 or log reduction values (LRVs) in TCID50

We look forward to receiving your revised manuscript.

Kind regards,

Caryn L Heldt, Ph.D.

Academic Editor

PLOS ONE

Journal Requirements:

4. We note you have included a table to which you do not refer in the text of your manuscript. Please ensure that you refer to Table 3 in your text; if accepted, production will need this reference to link the reader to the Table.

Reviewers' comments:

Reviewer's Responses to Questions

**Comments to the Author**

1. Is the manuscript technically sound, and do the data support the conclusions?

Reviewer #1: Yes

Reviewer #2: Partly

2. Has the statistical analysis been performed appropriately and rigorously? 

Reviewer #1: Yes

Reviewer #2: Yes

3. Have the authors made all data underlying the findings in their manuscript fully available?

Reviewer #1: Yes

Reviewer #2: Yes

4. Is the manuscript presented in an intelligible fashion and written in standard English?

Reviewer #1: Yes

Reviewer #2: Yes

5. Review Comments to the Author

Reviewer #1: This manuscript provides valuable information on the presence of genomic information for porcine viruses of concern in spray dried porcine plasma. The estimation of TCID50 based on standard curves is also of interest, as is the % positive sample information for SDPP from different geographical regions.

I have annotated the manuscript with suggestions for improving English language usage and reader understanding. Please use these suggestions if you agree. I have also asked some questions that can be addressed in your revision.

Reviewer #2: This is a well-written manuscript that contributes greatly to the literature and warrants publication. However, there is heavy discussion and conclusions that the estimated level of contamination was very low or low, whereas the data suggest otherwise. There was no prior establishment of what constitutes 'high' 'moderate' or 'low' levels of contamination, so these conclusions are entirely that of the authors, which have a conflict of interest in the data interpretation to be low. In reality, up to 100% contamination and relatively low Ct (below that of the established infectious dose) suggest there is risk in this ingredient, but the abstract, discussion, and conclusions try to convince the reader otherwise. It would be more appropriate to recognize there are a high number of positive samples with Ct below the known infectious dose, but that it is likely these are nonviable gene fragments that were inactivated during the manufacturing process.

6. PLOS authors have the option to publish the peer review history of their article (what does this mean?). If published, this will include your full peer review and any attached files.

Reviewer #1: **Yes: **Raymond W. Nims

Reviewer #2: No

---

## [Author Response · Author response to Decision Letter 0]

23 Feb 2022

Sorry we did not know how to include the funding information in the system. We add this information in the revised version of the manuscript

We apologized for the misunderstanding. In fact, we provide the data of all the information included in the manuscript. We clarified this subject in the revised version.

4. We note you have included a table to which you do not refer in the text of your manuscript. Please ensure that you refer to Table 3 in your text; if accepted, production will need this reference to link the reader to the Table.

The Table 3 was already included in the text of the initial submission. Lines 234-235 as Tables 2 and 3. In the new revised version appears in line 236-237 as Table 2 and Table3. 

Captions have been included for the Supporting information files. We have identified the supplementary tables as:

S1 Table -SDPP. Ct values and estimated virus genome presence in SDPP per months during the years 2018-2019.

S2 Table – Raw Plasma. Estimated virus genome presence in raw plasma per months during the years 2018-2019.

Reviewer’s Comments to the Author

Reviewer #1: This manuscript provides valuable information on the presence of genomic information for porcine viruses of concern in spray dried porcine plasma. The estimation of TCID50 based on standard curves is also of interest, as is the % positive sample information for SDPP from different geographical regions.

The authors appreciated the comments from Reviewer #1 because confirms our believe that the information provided in the manuscript can be of relevance for the swine sector, not only for understanding the presence of swine viral genome in raw or SDPP but also as a picture of the presence of different viruses in the different geographical areas studied. This information can be of importance for surveillance studies because provide quantitative information about the incidence of the different virus of concern for the swine industry.

I have annotated the manuscript with suggestions for improving English language usage and reader understanding. Please use these suggestions if you agree. I have also asked some questions that can be addressed in your revision.

The authors appreciated all the grammar suggestions provided by the Reviewer #1. We incorporated all of them in the revised manuscript.

Regarding the questions from reviewers #1. Find below the authors response to these questions.

Line # 55. “…multiple hurdles that…”. The reviewer is suggesting to change the word “hurdles” by “steps” throughout the manuscript.

To the authors, the term “multiple hurdles” is a common term to describe food manufacturing systems as can be found in the publication of Leistner (2000). Int. J. Food Microb. 

55:181-186. https://edisciplinas.usp.br/pluginfile.php/128773/mod_resource/content/1/Leistner.2000.pdf

Therefore, since this is a usual wording for the field being here studied, the authors prefer to keep the term “multiple hurdles” throughout the article.

Line #56. “…SD, 80oC throughout substance…”. How much time?

Throughout substance means that the particle achieves 80ºC during the drying process but the time that the particle is at that temperature is unknown and is variable depending on the driers design and capacity. The safety heat treatment is to achieve this 80ºC. It is the same safety concept that in case of meat for human consumption, the heat treatment at a minimum temperature of 80°C, which must be reached throughout the meat but without specifying the time at that temperature (European Council Directive 2002/99/EC. Annex III).

Line #56. “…ultraviolet light (UV) treatment (3000 J/L)…”. J/m2

No, the values are correctly expressed as J/L and not J/m2 as suggested by the reviewer #1. We refer to the paper of Blázquez et al., 2019. Reference [17] in the manuscript. In the case of treating liquid plasma, the UV system is a continuous flow reactor designed for exposing a liquid to UV light. Therefore, J/L is the appropriate unit of measure. 

Line #139. “…After 1.5 hours at 37ºC, inoculum was removed, and 30 mL of medium were added. Titration was done in triplicate obtaining a final titer of 105.48 TCID50/mL.”. Two significant figures is enough. The titration assay is not accurate to three significant figures, I think. 

The authors did not fully understand the comment from the reviewer #1. If he/she is referring that expressing the virus titer with maximum two decimals and not three as appear in some cases, we corrected this mistake in the revised version of the manuscript. 

We assume that some were expressed with three significant figures due to the effect of the mathematical mean. We apologize for the mistake and thank you for correcting it.

Line #186. “…A viral suspension was obtained and titrated in triplicate, obtaining a final viral solution of 106.64 TCID50 /mL. “ Provide incubation duration, is CPE the endpoint?

Sorry, we forgot to provide this information. Thank you for your observation. Once the flasks were inoculated, they were incubated at 37ºC for four days, at which time a clear CPE is observed and the infection was stopped.

We add this information on line 188 of the revised manuscript:

“After that time, the contents of the tube were transferred to a 175 cm2 flask, in which 40 mL of MEM-E supplemented with 1% pyruvate were added. Inoculated flasks were incubated for four days at 37ºC until CPE was observed. A viral suspension was obtained and titrated in triplicate, obtaining a final viral solution of 106.64 TCID50 /mL. “

Line #197. “… Each kit contained a genome quantified standard for the different vi 195 ruses tested: PRRSV (PRRSV-I dtec-RT-qPCR, PRRSV-II dtec-RT-qPCR), PEDV (PEDV dtec-RT-qPCR), PPV (PPV-1 dtec-RT-qPCR) and SIV (SIV dtec-RT-qPCR).” Add information for PCV?

We do not provide information about PCV-2 in this section since in the case of PCV-2 it was not necessary to perform a quantitative PCR again due that the first PCR performed (PCV-2 (LSI VetMAXTM Porcine Circovirus Type 2 Quantification, Thermo Fisher Scientific, MA, USA)) was already quantitative.

Line #247. “…effective identification of non-symptomatic animals probably contributed to the absence of SVA genome in the tested SDPP samples.”. How are non-symptomatic animals identified?

SVA has been associated with lameness and cutaneous vesicles, as well as increased mortality in the first weeks of age. Therefore, the absence of these clinical signs, at least the ones affecting finishers would be one of the arguments by which SVA might not be detected. On the other hand, animals at abattoir are inspected ante-mortem, and presumably only animals non-displaying clinical signs are sacrificed and used for human consumption.

The reviewer is right that non-symptomatic animals is difficult to identified. We changed this paragraph in the revised version of the manuscript as following

“Despite SVA infected animals have been sporadically detected on-farm and at abattoirs during ante-mortem inspection [33], effective identification of farm outbreaks and surveillance system in place probably contributed to the absence of SVA genome in the tested SDPP samples”

Line #291. “…In all regions, the estimated level of PCV2- was low (<2 log10 TCID50/g SDPP that corresponds to less than 5 virus particles per g of raw plasma), while PPV presence was slightly higher (<2.0 log10 TCID50/g liquid plasma).” This value is the same as that for PCV-2. Am I missing something?

The value for PCV-2 was indicated per g of SDPP while the value for PPV was refer to liquid plasma. In the revised version we put both values as g of SDPP. This sentence has been changed in the revised version of the manuscript as following:

“In all regions, the estimated level of PCV-2 was <2 log10 TCID50/g SDPP, while PPV presence was <3.0 log10 TCID50/g SDPP.”

Line #308. “…It is speculated that differences in stunning method, design of collection trough or slower line speed of abattoirs in Europe and Brazil compared to that in US and Canada may contribute to different levels of SIV contamination.” How?

Slower line speed of abattoirs in Europe and Brazil may affect to have higher residence time during the collection period and therefore more time for potential contamination of blood with SIV at the slaughter line.

We changed this paragraph in the revised version of the manuscript as following:

“…It is speculated that slower line speed of abattoirs in Europe and Brazil compared to that in US and Canada resulting in longer time for blood collection that may contribute to increased levels of SIV contamination.”

Line #338. “…The authors have read the journal's policy and the authors of this manuscript have the following competing interests: EB, CR, and JPolo are employed by APC Europe, S.L.U.” JPujols?

Dr. Joan Pujols is a researcher of IRTA-CReSA and his never has been employed of APC Europe, S.L.U. or APC companies.

Line #584. Table 3. The reviewer #1 is asking why the range of the values of PEDV, PPV and PPRS-US are negative. 

We appreciate the comment from reviewer #1. As indicated in the title of the Table 3, these values are expressed in log10 and therefore a negative log10 values do not mean that the virus particle is negative, only that is less than 1 particle/g of liquid plasma. For example, log10 -1.76 TCID50/ g liquid plasma in fact, means 0.017 TCID50/g liquid plasma

Reviewer #2: This is a well-written manuscript that contributes greatly to the literature and warrants publication. However, there is heavy discussion and conclusions that the estimated level of contamination was very low or low, whereas the data suggest otherwise. There was no prior establishment of what constitutes 'high' 'moderate' or 'low' levels of contamination, so these conclusions are entirely that of the authors, which have a conflict of interest in the data interpretation to be low. In reality, up to 100% contamination and relatively low Ct (below that of the established infectious dose) suggest there is risk in this ingredient, but the abstract, discussion, and conclusions try to convince the reader otherwise. It would be more appropriate to recognize there are a high number of positive samples with Ct below the known infectious dose, but that it is likely these are nonviable gene fragments that were inactivated during the manufacturing process.

We appreciated the comment received from the reviewer #2. It is true that the term high, moderate, or low levels of contamination is not defined and is subjective for each expert in the field. However, from the authors, having average values of log10 0.1 for PEDV, 1.56 for PCV-2, -1.4 for PRRSV US strain or 0.17 for PRRSV EU strain are relatively low, especially when compare with values of these viruses as the peak of viremia. Importantly, those values are considered low from the point of view of their potential infectiousness in an animal model. For example, 5.6 × 101 TCID50/g (1.75 log10 TCID50/g) was the minimum PEDV dose in feed to be infective for pigs (Schumacher et al., 2016, Am J Vet Res 77(10):1108-13). For PRRSV USA, a 2-mL inoculum containing 101 fluorescent foci units (equivalent to TCID50) of virus per milliliter was found sufficient to achieve infection by either route, but it was the very minimal dose to get it (Yoon et al., 1999 Vet Res 30(6):629-38). Therefore, all values obtained in plasma at slaughter should be considered within the low range of viral loads, most of them below the infectiousness levels as demonstrated when using one of the most susceptible routes of inoculation, the intraperitoneal one (Blázquez et al., 2019 Vet Microbiol 239:108450). Moreover, the likelihood of infection through the oral route is usually much lower compared to other routes such as drinking water or intranasal/oropharyngeal exposure as demonstrated with ASFV for example (Niederwerder et al., 2021 AnimaIs 11, 792). Therefore, all together would point out to low levels of contamination, also assuming this can be considered a subjective appraisal.

In any case, throughout the manuscript we have tried to eliminate this ambiguity of low or high levels of contamination. You can find these changes in the revised version of the manuscript.

Finally, although it is true that some co-authors have conflict of interest, all the data from the study is readily available in tables and supplementary material for the readers and it cannot be underestimated that other co-authors signing this manuscript are worldwide very well-recognized virology researchers in the field. Therefore, the authors consider that there is no bias with the information provided in this manuscript.

---

## [Editor Report · Decision Letter 1]

14 Mar 2022

PONE-D-21-33668R1Estimated quantity of swine virus genomes based on quantitative PCR analysis in spray-dried porcine plasma samples collected from multiple manufacturing plants.PLOS ONE

Dear Dr. Polo Pozo,

Thank you for submitting your manuscript to PLOS ONE. After careful consideration, we feel that it has merit but does not fully meet PLOS ONE’s publication criteria as it currently stands. Therefore, we invite you to submit a revised version of the manuscript that addresses the points raised during the review process.

Please see my comments below==============================

We look forward to receiving your revised manuscript.

Kind regards,

Caryn L Heldt, Ph.D.

Academic Editor

PLOS ONE

Journal Requirements:

Additional Editor Comments (if provided):

Thank you for addressing the reviewers comments. I ask that you fix the significant figures and add errors for your stock titers.

For all of the titers given in the methods, the significant figures are too high and errors are needed. Example, it says for PRRS (3268 EU strain) that the final titer was 10^5.48 TCID50/ml and done in triplicate. An error needs to be associated with the final titer and then the significant digits are so that there is only one significant digit in the error. So, if the error is 10^0.5, then the titer is 10^5.5 ± 10^0.5.

The error should also be fixed in Table 2 and Table 3. The error should have one significant digit and this will tell you how many digits you can use in the number reported. See https://www.ruf.rice.edu/~bioslabs/tools/data_analysis/errors_sigfigs.html. Error and significant figures need also be fixed in the supplemental data.
---

## [Author Response · Author response to Decision Letter 1]

23 Mar 2022

Additional Editor Comments (if provided):

Thank you for addressing the reviewers comments. I ask that you fix the significant figures and add errors for your stock titers.

For all of the titers given in the methods, the significant figures are too high and errors are needed. Example, it says for PRRS (3268 EU strain) that the final titer was 10^5.48 TCID50/ml and done in triplicate. An error needs to be associated with the final titer and then the significant digits are so that there is only one significant digit in the error. So, if the error is 10^0.5, then the titer is 10^5.5 ± 10^0.5.

The authors appreciated the suggestion from the Academic Editor. We have included in the new version of the manuscript the error for the titers in the inoculum. As example the new reported information for the PRRS EU strain (3268EU) inoculum is as follow: ”… final titer of 105.5±0.2 TCID50/mL.”

The error should also be fixed in Table 2 and Table 3. The error should have one significant digit and this will tell you how many digits you can use in the number reported. See https://www.ruf.rice.edu/~bioslabs/tools/data_analysis/errors_sigfigs.html. Error and significant figures need also be fixed in the supplemental data.

The authors really appreciated the help from the Academic Editor to clarify the request and provide the significant digit for all data included in Table2 and Table 3 and Supplementary Table 1 and Suplementary Table 2. We are deeply in debt for her willing to review all the data in advance to the new submission. 

The new revised version of the manuscript includes the correct format for all the data in the mentioned tables.

---

## [Editor Report · Decision Letter 2]

6 Apr 2022

Estimated quantity of swine virus genomes based on quantitative PCR analysis in spray-dried porcine plasma samples collected from multiple manufacturing plants.

PONE-D-21-33668R2

Dear Dr. Polo Pozo,

We’re pleased to inform you that your manuscript has been judged scientifically suitable for publication and will be formally accepted for publication once it meets all outstanding technical requirements. Thank you for all of your work on the error analysis.

Kind regards,

Caryn L Heldt, Ph.D.

Academic Editor

PLOS ONE
---

## [Editor Report · Acceptance letter]

11 Apr 2022

PONE-D-21-33668R2 

Estimated quantity of swine virus genomes based on quantitative PCR analysis in spray-dried porcine plasma samples collected from multiple manufacturing plants. 

Dear Dr. Polo Pozo:

I'm pleased to inform you that your manuscript has been deemed suitable for publication in PLOS ONE. Congratulations! Your manuscript is now with our production department. 

Kind regards, 

on behalf of

Dr. Caryn L Heldt 

Academic Editor

PLOS ONE